# May Mangafodipir or Other SOD Mimetics Contribute to Better Care in COVID-19 Patients?

**DOI:** 10.3390/antiox9100971

**Published:** 2020-10-10

**Authors:** Jan Olof G Karlsson, Per Jynge, Louis J Ignarro

**Affiliations:** 1Division of Drug Research/Pharmacology, Linköping University, SE-581 83 Linköping, Sweden; 2Innlandet Trust Hospital, Gjøvik Hospital, NO-2819 Gjøvik, Norway; per.jynge.ha@gmail.com; 3Department of Pharmacology, UCLA School of Medicine, 264 El Camino Drive, Beverly Hills, CA 90212, USA; lignarro@gmail.com

**Keywords:** mangafodipir, inflammation, SARS-CoV-2, COVID-19, MnSOD-mimetic, nitric oxide, superoxide

## Abstract

Severe acute respiratory syndrome coronavirus 2 (SARS-CoV-2) is characterized by massive inflammation of the arterial endothelium accompanied by vasoconstriction and widespread pulmonary micro thrombi. As a result, due to the destruction of nitric oxide (^•^NO) by inflammatory superoxide (O_2_^•−^), pulmonary ^•^NO concentration ceases, resulting in uncontrolled platelet aggregation and massive thrombosis, which kills the patients. Introducing ^•^NO by inhalation (INO) may replace the loss of endothelium-derived ^•^NO. The first results from clinical trials with INO in SARS-CoV-2 patients show a rapid and sustained improvement in cardiopulmonary function and decreased inflammation. An ongoing phase III study is expected to confirm the method’s efficacy. INO may hence become a first line treatment in SARS-CoV-2 patients. However, due to the rapid inactivation of ^•^NO by deoxyhemoglobin to nitrate, pulmonary administration of ^•^NO will not protect remote organs. Another INO-related pharmacological approach to protect SARS-CoV-2 patients from developing life-threatening disease is to inhibit the O_2_^•−^-driven destruction of ^•^NO by neutralizing inflammatory O_2_^•−^. By making use of low molecular weight compounds that mimic the action of the enzyme manganese superoxide dismutase (MnSOD). The MnSOD mimetics of the so-called porphyrin type (e.g., AEOL 10150), salen type (e.g., EUK-8) and cyclic polyamine type (e.g., M40419, today known as GC4419 and avasopasem manganese) have all been shown to positively affect the inflammatory response in lung epithelial cells in preclinical models of chronic obstructive pulmonary disease. The Manganese diPyridoxyL EthylDiamine (MnPLED)-type mangafodipir (manganese dipyridoxyl diphosphate—MnDPDP), a magnetic resonance imaging (MRI) contrast agent that possesses MnSOD mimetic activity, has shown promising results in various forms of inflammation, in preclinical as well as clinical settings. Intravenously administration of mangafodipir will, in contrast to INO, reach remote organs and may hence become an important supplement to INO. From the authors’ viewpoint, it appears logical to test mangafodipr in COVID-19 patients at risk of developing life-threatening SARS-CoV-2. Five days after submission of the current manuscript, Galera Pharmaceuticals Inc. announced the dosing of the first patient in a randomized, double-blind pilot phase II clinical trial with GC4419 for COVID-19. The study was first posted on ClinicalTrials.gov (Identifier: NCT04555096) 18 September 2020.

Johan Giesecke, a leading expert on the epidemiology of infectious diseases and a member of the WHO Strategic and Technical Advisory Group for Infectious Hazards recently concluded [1]: “Everyone will be exposed to severe acute respiratory syndrome coronavirus 2 (SARS-CoV-2), and most people will become infected; the disease is known as COVID-19. The virus is spreading like wildfire in all countries; but we do not see it - it almost always spreads from younger people with no or weak symptoms” … “There is very little we can do to prevent this spread: a lockdown might delay severe cases for a while; but once restrictions are eased; cases will reappear”. 

Effective drugs that save lives might soon be developed, but this pandemic is spreading extremely fast, and those drugs have to be developed, tested, and marketed at a tremendously high speed. Much hope is put into vaccines, but they will take time, and with the unclear protective immunological response to infection, it is uncertain if vaccines will be very effective [1]. Researchers are hence looking for drugs already approved for other indications that might be efficacious towards SARS-CoV-2. 

SARS-CoV-2 is driven by the human immune system itself by a violent inflammatory response to the viral intruder. The Chinese Center for Disease Control and Prevention recently published the largest case series to date of coronavirus disease 2019 (COVID-19) in mainland China [2]. Most cases were classified as mild, but in 5% of the cases they were critical, with severe pneumonia, respiratory failure, septic shock, and/or multiple organ dysfunction or failure. Except for standard intensive care, including invasive ventilation, there are, at present, few other options. Preliminary results from an open-label study in patients hospitalized with SARS-CoV-2 displayed a lower mortality among those who were receiving invasive ventilatory support and dexamethasone compared to those that did not receive dexamethasone, 29.3% vs. 41.4%, respectively [3], suggesting that standard intensive care should include treatment with glucocorticoids. However, such a measure will far from save all lives.

A rather natural approach would be to look for other drugs that, like glucocorticoids, lower the inflammatory burden. Such an approach apparently demands drugs targeting central mechanisms that can turn off the inflammatory storm. SARS-CoV-2 is characterized by massive inflammation of the arterial endothelium, accompanied by vasoconstriction and widespread pulmonary micro thrombi. As a result, due to the destruction of nitric oxide (^•^NO) by inflammatory superoxide (O_2_^•−^), pulmonary ^•^NO concentration ceases, resulting in uncontrolled platelet aggregation and massive thrombosis, which kills the patient. Introducing ^•^NO by inhalation (INO) may replace the loss of endothelium-derived ^•^NO. Relevant situations to compare are those of inhaled INO in newborn babies with persistent pulmonary hypertension (PPHN) and in seriously ill SARS CoV-1 patients during the outbreak in China 2004 [4]. Inhalation of very small amounts INO gas by newborn babies with life-threatening PPHN results in a dramatic and permanent reversal of pulmonary vasoconstriction. INO literally turns blue babies into pink babies [5]. Regarding SARS-CoV-1, administration of INO reversed pulmonary hypertension, improved severe hypoxia, and shortened the length of ventilatory support compared to matched control patients with SARS-CoV-1 not receiving treatment [4]. These promising results seen in SARS-CoV-1 patients after INO have led to several ongoing clinical trials with INO in SARS-CoV-2 patients, including a large phase III study planned to include 500 patients (ClinicalTrials.gov Identifier: NCT04421508). Results from a study including six severely ill pregnant COVID-19 patients receiving INO admitted to the Massachusetts General Hospital have very recently been reported [6]. The results show a rapid and sustained improvement in cardiopulmonary function and decreased inflammation. 

Another INO-related pharmacological approach to protect COVID-19 patients from developing life-threatening disease is to inhibit the superoxide (O_2_^•−^)-driven destruction of ^•^NO. It is well known that the common key players in inflammatory responses caused by various pathological conditions are ^•^NO, O_2_^•−^ and peroxynitrite (ONOO^−^), and they have been characterized by Beckman and Koppenol as “the good”, “the bad”, “and ugly”, respectively [7,8]. Superoxide readily reacts with ^•^NO to form ONOO^−^, revealing the “ugly” side. Under normal cellular conditions, this reaction is kept in check mainly by the mitochondrial manganese-containing superoxide dismutase (MnSOD) [9]. However, under conditions of high inflammatory stress, the amount of O_2_^•−^ rises to levels exceeding the enzymatic capacity of MnSOD, leading to destruction of “good” ^•^NO and the production of “ugly” ONOO^−^. This will, in turn, cause a redox metal-driven nitration of tyrosine residues, including tyrosine 34 and tyrosine 74 of the human MnSOD and cytochrome c, respectively. Nitration of tyrosine 34 results in an irreversible inhibition of MnSOD that severely aggravates inflammation. Hence, pharmacological strategies aiming to preserve endogenous ^•^NO in COVID-19 patients at risk appear as a logical possibility. Such a strategy could be achieved by making use of low molecular weight compounds with catalytic MnSOD activity, i.e., so-called MnSOD mimetics. The MnSOD mimetics of the so-called porphyrin type (e.g., AEOL 10150), salen type (e.g., EUK-8) and cyclic polyamine type (e.g., M40419, today known as GC4419 and avasopasem manganese) have all been shown to positively affect the inflammatory response in lung epithelial cells in preclinical models of chronic obstructive pulmonary disease [10]. Mangafodipir (a magnetic resonance imaging (MRI) contrast agent), which possesses MnSOD mimetic activity, has shown promising results in various forms of inflammation, in preclinical as well as clinical settings [9]. Importantly, due the rapid inactivation of ^•^NO in the blood by deoxyhemoglobin to nitrate, pulmonary (or intravenous) administration of ^•^NO will not protect remote organs. Intravenous administration of an MnSOD mimetic will, on the other hand, readily reach remote organs, and may hence become an important supplement to INO in patients at risk of developing multiple organ failure. 

More than 25 years ago, during the clinical development of mangafodipir as a magnetic resonance imaging (MRI) contrast agent, it was serendipitously discovered that this compound and its metabolite manganese diPyridoxyL EthylDiamine (MnPLED) possessed MnSOD mimetic activity [9]. Rapid intravenous administration in patients caused vasodilation and facial flushing. Classical organ bath experiments suggested that these compounds possessed MnSOD mimetic activity and hence protected endothelial ^•^NO from reacting with O_2_^•−^, consequently preserving vasodilatory ^•^NO. The MnSOD mimetic activity was subsequently confirmed by electron paramagnetic resonance (EPR) spectroscopy [9]. In addition, these compounds knockout redox-active metal ions, such as Fe^2+^/Fe^3+^/Fe^4+^, which are necessary for causing ONOO^−^-mediated tyrosine nitration, because they have a tremendously high affinity for these metal ions. Preclinical studies with mangafodipir have shown convincing therapeutic effects in various pathological conditions of acute inflammation, including myocardial infarction [9], paracetamol (acetaminophen)-induced liver failure [11] and chemotherapy-induced injuries of normal tissue, including doxorubicin-induced heart failure [9] and paclitaxel-induced fibril neutropenia [12]. The paracetamol study [11] deserves some extra attention, as it demonstrated the impressive curative efficacy of mangafodipir in mice when administered 6 hours after intoxication with a lethal dose of acetaminophen, at a time point where standard treatment with N-acetylcysteine was without effect. Feasibility phase II studies suggest that the above described therapeutic effects also are present in humans [9]. Of particular interest is a recent phase I/II study in paracetamol-intoxicated patients where the addition of the mangafodipir analog calmangafodipir was well tolerated when combined with N-acetylcysteine and where preliminary efficacy data suggest that it may reduce paracetamol liver toxicity [13].

The proposed protective mechanisms of mangafodipir are as follows: (i) it lowers the intracellular level of O_2_^•−^ directly through its MnSOD mimetic activity and preserves endothelium-derived ^•^NO; (ii) it binds and disarms redox-active metals and thus arrests the nitration of tyrosine 34 and tyrosine 74 of mitochondrial MnSOD and cytochrome c, respectively; and (iii) it replaces endogenous mitochondrial MnSOD that has been irreversibly inactivated. Mangafodipir and analogs may hence, through their combined MnSOD mimetic activity and binding capacity for redox-active metals, lower the inflammatory burden in critical SARS-CoV-2 infections to a level where the survival rate increases considerably. Due to the above described rapid inactivation of ^•^NO by deoxyhemoglobin to nitrate, pulmonary administration of ^•^NO will not protect remote organs. On the other hand, intravenous administration of managfodipir will reach remote organs, and may hence become an important supplement to INO, particularly in patients at risk of developing multiple organ failure. From our viewpoint, it appears important to test mangafodipir in COVID-19 patients at risk of developing life-threatening SARS-CoV-2.

Five days after the submission of the current manuscript to *Antioxidants*, Galera Pharmaceuticals Inc. announced the dosing of the first patient in a randomized, double-blind pilot phase II clinical trial with the MnSOD mimetic GC4419 for COVID-19, first posted on ClinicalTrials.gov (Identifier: NCT04555096) 18 September 2020. Thus, the path may be “paved” for other MnSOD mimetics, including mangafodipir, to enter the clinical COVID-19 arena. When it comes to mangafodipir, its combined MnSOD mimetic and redox metal binding activity may be of particular importance when attacking the inflammatory storm in SARS-CoV-2 patients. It may also be of great importance that mangafodipir is a contrast medium with a safe record in clinical use over more than one decade.

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
