# Peer review of "May Mangafodipir or Other SOD Mimetics Contribute to Better Care in COVID-19 Patients?"

_antioxidants, 2020, doi:10.3390/antiox9100971_

Round 1

Reviewer 1 Report

The authors have submitted a "viewpoint" article proposing the use of an SOD mimetic to stabilize NO in the pulmonary microcirculation as a treatment for COVID-19. Journals are currently being innundated with submissions regarding the pathophysiology and, in particular, potential treatments for COVID-19. In most cases, these reports add more to the random background noise rather than providing clarity. While a viewpoint discussing the basis for stabilizing NO to prevent pulmonary injury as well as potential mechanisms for delivering or stabilizing NO might have value, the focus on a single treatment in which the authors have significant financial interest while providing no directly relevant data (ie, from COVID-19 patients) greatly diminishes the value.

Author Response

The authors agree that the focus on a single treatment with mangafodipir may appear unfortunate. The manuscript has therefore been extended with other MnSOD-mimetics, of the so-called cyclic polyamine, porphyrin and salin types. A relevant review by Rahman and Adcock 2006 has been included in the reference list (a reference that has been cited more than 900 times). The Galera Therapeutics Inc news that  the first COVID-19 patient has been dosed with their GC4419, a cyclic polyamine type of compound, appeared 5 days after the current manuscript submission, apparently indicating that MnSOD-mimetics add more than “random background noise”. This important news has been included in the manuscript. When it comes to authors’ financial interest, it should be stressed that the overwhelming majority of all useful drugs are the result of commercially driven developmental processes. Today that also includes the development of vaccines. To conduct clinical studies is costly and in this particular case there is an urgent need of effective medicines. Together with colleges the authors founded FODI SCIENCE AB in order to attract venture capital enabling fast-track clinical studies in COVID-19 patients. This way of developing new medicines, in this case a new indication of a drug, previously approved for diagnostic imaging, is a standardized procedure for start-up pharmaceutical companies. As long as the authors declare their conflicts of interest that should be rather unproblematic.       

Reviewer 2 Report

The manuscript describes a new potential therapeutic strategy to counteract the pulmonary damage caused by SARS-Cov-2 infection. It is based on the use of some SOD-mimetic compounds, which, indirectly, can increase the endogenous NO availability. The assumption is of interest and the data reported to support this therapeutic strategy are suitable.

The manuscript is worthy of publication in the present form.

Author Response

The manuscript is, according to Reviewer 2, "worthy of publication in the present form" 

Reviewer 3 Report

This is a very provocative viewpoint article from Karlsson and colleagues who argue for the use of mNSOD mimetic for treating the remote organ damage associated with SARS-CoV-2.  This is a subject matter that is of wide interest to teh world population at the moment and is worthy of publication.  There are only a few minor grammar issues that need to ebb ddressed

  1.  Line 20: Another but INO-related... Please clarify meaning.

2.  Line 26: will in opposite INO...do the authors mean in contrast to INO?

3. Line 73.  Please rewrite the sentence beginning " Another but INO-related..." for clarity its unclear what the authors are indicating.

4. Line 87: "due rapid".   Should this be "due to the rapid"?

5. Line 89: intravenously administration... should be intravenous.

6. Line 90 "to INO in patient" should be in patients.

7. Line 100: tremendous should be tremendously.

8. Line 123: INO in patient should be INO in patients.

9. Line 124: "seems highly motivated to test".  Please rewrite for clarity. Its unclear what the authors meaning is.

Author Response

The manuscript has been revised in accordance with the comments by Reviewer 3 

Round 2

Reviewer 1 Report

The authors have revised the manuscript by broadening the scope of the paper to include other Mn-SOD mimetics including one that recently entered clinical trials. This does change my view of the report. This reviewer acknowledges that without commercialization, new therapies never enter clinical use. Moreover, those with a financial interest in development of a therapy are often the most qualified to study its efficacy. My objection to the original submission was that it was essentially an opinion piece focused on one drug. By placing it into context of other similar drugs being developed, the authors have largely mitigated my objections.